# Cladoceran *Chydorus sphaericus* and Colonial Cyanobacteria: Potentially a Toxic Relationship?

**DOI:** 10.3390/toxins17060298

**Published:** 2025-06-12

**Authors:** Helen Agasild, Ilmar Tõnno, Margarita E. Gonzales Ferraz, Peeter Nõges, Priit Zingel, Lea Tuvikene, René Freiberg, Tiina Nõges, Kristel Panksep

**Affiliations:** 1Institute of Agricultural and Environmental Sciences, Estonian University of Life Sciences, 51006 Tartu, Estonia; ilmar.tonno@emu.ee (I.T.); margarita.gonzales@emu.ee (M.E.G.F.); peeter.noges@emu.ee (P.N.); priit.zingel@emu.ee (P.Z.); lea.tuvikene@emu.ee (L.T.); rene.freiberg@emu.ee (R.F.); tiina.noges@emu.ee (T.N.); kristel.panksep@ut.ee (K.P.); 2Institute of Technology, University of Tartu, 50411 Tartu, Estonia; 3Department of Aquatic Resources, Swedish University of Agricultural Sciences, Ulls gränd 1, SE-756 51 Uppsala, Sweden

**Keywords:** *Chydorus sphaericus*, *Microcystis*, aquatic food web, in situ feeding

## Abstract

*Chydorus sphaericus* is often a dominant cladoceran zooplankton species in water bodies experiencing harmful cyanobacterial blooms. However, its relationship with toxin-producing algae remains largely unexplored. In this study, the feeding behavior of *C. sphaericus* on colonial cyanobacteria and potentially toxic *Microcystis* was investigated in a temperate, shallow, eutrophic lake. Liquid chromatographic analyses of phytoplankton marker pigments in *C. sphaericus* gut content revealed that pigments characteristic of cyanobacteria (identified a zeaxanthin, echinenone, and canthaxanthin) comprised the majority of its diet. Among them, colonial cyanobacteria (marked by the pigment canthaxanthin) were the highly preferred food source despite their minor contribution to phytoplankton biomass. qPCR targeting *Microcystis* genus-specific *mcyE* synthase genes, which are involved in microcystin biosynthesis, indicated that potentially toxic strains of *Microcystis* were present in *C. sphaericus* gut content throughout its temporal and spatial presence in the lake. The results suggest that the common small cladoceran in eutrophic waters, *C. sphaericus*, has a close trophic interaction with colonial cyanobacteria (including *Microcystis*) and may represent an important vector for transferring toxigenic *Microcystis* to the food web, even under conditions of low *Microcystis* biomass in the lake water.

## 1. Introduction

In recent decades, toxic cyanobacterial blooms in freshwater systems worldwide have increased in both frequency and severity [1]. Colonial cyanobacteria, such as *Microcystis* spp., are often among the genera that form the harmful algal blooms [2]. These cyanobacteria form large gelatinous colonies composed of small coccoid cells. The aggregates, often larger than 50 µm, are generally inedible or poorly edible due to their size, interference with feeding, and the production of toxic metabolites, such as microcystins [3,4]. Although this type of grazing resistance is rather effective, zooplankton have developed strategies to overcome these negative effects, such as smaller size, selective feeding, and resistance to toxicity, reviewed by Moustaka-Gouni and Sommer [5]. Despite their potential toxicity, cyanobacteria often represent the largest pool of algal biomass in eutrophic waters and serve as food for various zooplankton, including protozoa, such as *Paramaecium* sp. [6,7], rotifers [8], cladocerans [9], and both calanoid and cyclopoid copepods [10,11]. These grazers differ in their feeding rates, ability to select for cyanobacteria [12,13], and survival and fecundity on potentially toxic cyanobacteria [14], leading to variable interactions with this phytoplankton.

Furthermore, recent research [15] has evidenced that copepod gut microbiota (*Acartia bifilosa* and *Eurytemora affinis*) can effectively break down microcystins, providing an adaptive mechanism for feeding on toxin-producing cyanobacteria during blooms. While cyanobacterial grazing is often viewed as a potential biological control measure against toxic cyanobacteria, in aquatic ecosystems, the ingested toxic cells are simply ‘packed and ready’ for zooplanktivorous predators (predatory zooplankton and fish), potentially causing further contamination problems in food chains. Therefore, to understand the complex interactions between toxin-producing cyanobacteria and zooplankton, further research on grazer species co-occurring with cyanobacterial blooms is essential.

Regarding feeding and survival on cyanobacteria, cladocerans are among the most studied zooplankters, particularly the planktonic genus *Daphnia* [16,17]. Research has shown that the large, generalist-feeding *Daphnia* can be severely impaired by potentially toxic cyanobacteria and tend to decrease or disappear with deteriorating water quality due to eutrophication and bloom formation [18,19,20]. Relatively less attention, however, has been paid to species that co-occur or increase in abundance during cyanobacterial blooms, such as *C. sphaericus*. *C. sphaericus* is one of the small-sized cladocerans that often appears as a common plankter in eutrophic waters in which extensive cyanobacteria blooms are prevalent [21,22]. Unlike other chydorid species inhabiting littoral zones, it is well adapted to a planktonic lifestyle, using mats or filaments of algae, including cyanobacteria, as its substratum [23,24].

Analyses of numerous lakes across the gradient of trophic states in Europe and the USA have indicated that the dominance of *C. sphaericus* is linked to eutrophication and higher levels of chlorophyll *a* [25,26,27]. The abundance, dominance among zooplankton, and body size of *C. sphaericus* are suggested as good bioindicators for environmental conditions and trophic status [26,28,29]. Although *C. sphaericus* remains rare in the tropics and subtropics of the Eastern Hemisphere, its anthropogenic introduction—primarily through the stocking of fish from Europe—is facilitating its dispersal across the region [30,31]. It is assumed that rather than being a single species, *C. sphaericus sensu lato* (*s.l.*) consists of a complex of taxa, including at least three potentially valid species: *C. sphaericus s.str*., *Chydorus biovatus* Frey, 1985, and *Chydorus brevilabris* Frey, 1980 [32,33]. Since species differentiation is not the focus of this study, we refer to the specimens studied in this research as *C. sphaericus.*

Recent research has shown that zooplankton clones constantly exposed to cyanobacteria develop genotypic resistance to toxins and exhibit better growth in cyanobacterial environments [34]. This applies to *C. sphaericus* as well. Living in environments with toxic cyanobacteria, *C. sphaericus* has evolved a tolerance to these organisms, which is further enhanced with increasing microcystin concentrations in the environment [35]. Despite *C. sphaericus*’s prevalence in bloom-forming waterbodies, its direct feeding interactions with cyanobacteria have been seldom investigated [22]. Research indicates that *C. sphaericus* is well adapted to the cyanobacterial detrital food. Cyanobacterial detritus added to green algae in laboratory culture experiments enhanced the growth and production of *C. sphaericus* [22]. However, the authors of another study found that *C. sphaericus* fed with poor-quality food (50% fresh *Microcystis aeruginosa* and 50% *Chlorella pyrenoidosa*) had significantly lower population growth than the conspecifics fed good-quality food (100% *C. pyrenoidosa*) [35]. Tõnno and others [36] investigated crustacean zooplankton algal diet selection in a eutrophic Lake Võrtsjärv (Estonia) and found that phytoplankton pigments from colonial cyanobacteria (most probably *Microcystis* spp.) constituted a major source of algal food for *C. sphaericus.* However, only three samples of *C. sphaericus* were analyzed for feeding assessment over its seasonal presence, leaving an open question about its relationship with potentially toxic *Microcystis* [36].

We conducted further research using high-performance liquid chromatography (HPLC) and quantitative PCR (qPCR) methods to more precisely assess the relationship between colonial cyanobacteria and potentially toxic *Microcystis* with *C. sphaericus* in Lake Võrtsjärv (Figure 1). In this lake, *C. sphaericus* is the most abundant cladoceran species, often reaching numbers of 100 individuals per liter of water during the growing period [37]. Recent research using long-term data from this lake indicate that ongoing rising trends in temperature and cyanobacterial biomass are further favouring the increase in *C. sphaericus* abundance in this lake [37,38]. Although *Microcystis* represents a minor phytoplankton group in Võrtsjärv [36], it is the main microcystin producer there (K. Panksep, unpublished data). Based on this knowledge, we hypothesized that (1) colonial cyanobacteria form the main algal food source for *C. sphaericus* through its seasonal presence; (2) toxigenic *Microcystis* spp. contribute substantially to the diet of *C. sphaericus*; and (3) the consumption of toxic cells by *C. sphaericus* is positively associated with the biomass of *Microcystis* and the concentration of *Microcystis mcyE* gene copy numbers in lake water. To achieve this goal, we investigated the seasonal feeding of *C. sphaericus*, and compared the selectivity and ingestion of potentially toxic cyanobacteria across the lake area with variable presence of colonial cyanobacteria.

## 2. Results

### 2.1. Temperature, Phytoplankton, and C. sphaericus Seasonal Dynamics

In 2015, the ice cover melted by 22 March, followed by a steady increase in temperature up to 19.4 °C in August, and then decreased towards the end of December when the new ice cover formed (Figure 2). The water quality parameters are shown in Appendix A.

Phytoplankton biomass peaked in May and again in July–August during the highest temperatures, dominated by diatoms (*Aulacoseira* sp.) and filamentous cyanobacteria (mainly *Limnothrix planktonica* and *L. redekei*), respectively (Figure 3A). In August, the highest phytoplankton biomass across the lake occurred at sites 5 and 7 (Figure 3B), dominated by *L. planktonica* and *Aulacoseira* sp. The biomass of colonial cyanobacteria fluctuated seasonally between 0.035 and 2.495 mg WW/L, peaking in June and forming 11% of the total phytoplankton biomass. In August, colonial cyanobacteria had the highest biomasses at sampling site 2 (1.842 mg WW/L), site 4 (2.250 mg WW/L), and site 5 (2.018 mg WW/L). *Microcystis* spp. (*M. wesembergii*, *M. viridis*, *M. pulverea*, *Microcystis* sp.) and *Cyanodicton* sp. were the major taxa of colonial cyanobacteria, with minor contributions from *Gomphosphaeria lacustris*, *Gloeocapsa* spp. and *Merismopedia* spp. (Figure 3C,D).

The numbers of *C. sphaericus* fluctuated between 2 and 420 ind/L, with biomass ranging from 0.01 to 2.02 mg WW/L, peaking in June. In August, the highest abundance was observed at sites 7 and 6 (Figure 3E,F). Seasonally, *C. sphaericus* often dominated, contributing up to 79% of the total multicellular zooplankton biomass, although it generally contributed less than 30% in August across the lake.

### 2.2. Phytoplankton Marker Pigment Concentrations in Lake Water; C. sphaericus Feeding and Selectivity for Cyanobacteria

In lake water, the marker pigment compositions and dynamics followed the phytoplankton community structure (Figure 4A,B). Pigments generally characteristic of cyanobacteria (zeaxanthin and echinenone, belonging to both filamentous and colonial forms) commonly formed half of the pigment concentration both seasonally and spatially. The second largest group of the marker pigments (fucoxanhin and diadinoxanthin+diatoxanthin) belonged to diatoms. Marker pigments characteristic of green algae and cryptophytes made minor contributions.

In all studied samples, the gut content of *C. sphaericus* predominantly contained marker pigments Cantha (colonial cyanobacteria) and Echin (cyanobacteria), with a lesser amount of another cyanobacterial pigment, Zea (Figure 4C,D). Pigments related to cyanobacteria formed approximately 80% of the ingested algal pigments. Marker pigments of chlorophyta (Chl *b*, Lut, and Neox) were also detected in *C. sphaericus* guts, along with minor contributions from marker pigments of cryptophyta (Allo) and diatoms (Fuco and Diadino+Diato). Chesson’s selectivity index demonstrated a clear preference of Cantha and to a lesser extent, Echin in *C. sphaericus* diet throughout the investigated timespan and sampling sites (Figure 5).

### 2.3. Potentially Microcystin-Producing Cyanobacteria in Lake Water and in C. sphaericus Gut Content

The presence of *mcyE* genes in all studied water samples indicated potential microcystin production throughout the seasons and across the study area in August (Figure 6A,B; samples were not collected in January and June). The *Microcystis* genus was the predominant potential producer of the microcystin in Võrtsjärv. The measured abundance of *Microcystis mcyE* genes fluctuated between 342 and 3000/mL, peaking seasonally in March, July, and September and spatially in August at site P5 (Figure 6A,B).

Toxigenic *Microcystis* cells were present in the gut content of *C. sphaericus* on almost all dates and at almost all sites analyzed (Figure 6C,D). The abundance of toxigenic cells fluctuated between 1.1 and 7.4 cells per individual over the seasonal and spatial range (August). The analyses conducted in September and December for *C. sphaericus* samples were unsuccessful due to an insufficient amount of DNA gained from extraction. This limitation hindered the ability to obtain reliable results from these samples.

As there were no statistical differences in *Microcystis mcyE* abundances in lake water and *mcyE*-containing cells in *Chydorus* gut content between the sampling sites (Kruskal-Wallis test, *p* > 0.05), the data from all eight sites across the lake (1, 2, 3, 4, 5, 6, 7, 10) were analyzed together. Spearman’s rank correlation (*r*_s_) analysis did not reveal any significant (*p* > 0.05) relationships between *mcyE* copy numbers in lake water and phytoplankton, zooplankton (*C. sphaericus,* cladocerans, and total zooplankton), or most physicochemicalcharacteristics, except for temperature. The abundance of *Microcystis* toxic genotypes in lake water was negatively correlated with temperature (*r_s_* −0.5083, *p* < 0.05). The abundance of *Microcystis mcyE*-containing cells in *Chydorus* gut content had no statistically significant correlations with the tested parameters.

## 3. Discussion

*Chydorus sphaericus* is a small cladoceran that is often present with high abundance in eutrophic lakes with cyanobacterial dominance or toxigenic blooms [22,28,29]. Our results further complement this knowledge by explaining its close connection with cyanobacteria. We showed that *C. sphaericus* not only uses the filaments and colonies as a substratum [23,24] but also actively feeds on these algae. Phytoplankton marker pigment analysis indicated that *C. sphaericus* had a predominantly cyanobacteria-based diet, with approximately equal contributions from filamentous and colonial forms but a clear preference for colonial cyanobacteria. The active feeding pattern and selection for colonial cyanobacteria persisted throughout the lake and across seasons, even during periods of low water temperatures and ice cover. This confirms that this small cladoceran has a tighter connection to cyanobacteria, specifically colonial forms, than other co-occurring pelagic copepods and cladocerans, as revealed by earlier zooplankton gut pigment analyses in Võrtsjärv [36].

The strong dietary affinity of *C. sphaericus* towards colonial cyanobacteria is surprising, given that these algae form only a minor fraction of the phytoplankton biomass in Võrtsjärv. The average contribution of colonial cyanobacteria to total phytoplankton biomass is approximately 10% [36]. Compared to other common planktonic cladocerans, *Chydorus*’ dual feeding mode, combining filter and raptorial feeding, allows it to exploit both suspended seston particles and attached food sources, such as periphyton or algal colony cells [39]. *C. sphaericus* can use the setae of its second trunk limbs to scrape surface cells [24]. This likely explains the appreciable amount of pigments from colonial cyanobacteria in *C. sphaericus* gut content. Experiments with polystyrene fluorescent particles [40,41] have revealed that *C. sphaericus* is an efficient feeder on small particles, corresponding to the size of bacteria and small algal cells (such as individual cells of *Microcystis*, *Cyanodictyon*, etc.). Therefore, this may be a behavioural response to searching for suitable-sized food from the surfaces of algal colonies, which in turn supports its feeding on potentially toxic colonial cyanobacteria, such as *Microcystis*. We cannot detect a preference for certain colonial cyanobacteria based on current analysis methods. However, qPCR detection of *Microcystis*-specific *mcyE* genes clearly evidenced *C. sphaericus* ingestion of toxigenic *Microcystis*.

The occurrence of the *Microcystis mcyE* gene in Võrtsjärv persisted throughout the seasons, even under low water temperatures and low *Microcystis* biomass. This is in line with recent findings that winter populations of *Microcystis* are capable of synthesizing microcystins even during long-lasting ice cover in a permafrost lake [42]. Year-round assessments of *Microcystis* growth and microcystin-producing gene detection have rarely been conducted in temperate zones with seasonal ice formation. We studied the period from February to December, with water temperature ranging from 1.0 to 19.4 °C, and found a negative relationship between temperature and *mcyE* gene abundances. The relationships between microcystin-producing genotypes (*mcyA*, *mcyB*, *mcyD*, *mcyE*, and *mcyJ*) and temperature have been analyzed by several authors, with them finding positive association for all of these genes [43], and specifically for *mcyA* and *mcyE* [44], *mcyD* [45], *mcyB* and *mcyE* [46]; negative associations for *mcyD* [47]; or no association for *mcyB* [48,49] and *mcyE* [50,51]. This suggests that the potential to produce microcystin could be achieved under a variable combination of environmental conditions [48,49]. Our results suggest that low temperatures do not suppress microcystin synthetase gene production and that microcystins are most likely present year-round in Võrtsjärv water.

Concurrently, qPCR analysis confirmed that *C. sphaericus’s* diet included toxigenic *Microcystis* even during the cold period and across the lake area in August. However, contrary to predictions, the consumption of toxigenic cells by *C. sphaericus* was not correlated with the biomass of *Microcystis* nor the abundance of *Microcystis mcyE* gene copy numbers in lake water. Due to the scarcity of samples in the present study, further investigations are necessary to confirm this feeding pattern. The average ingestion per individual was 4.2 cells, representing the last feeding before sampling. Similar individual ingestion of *C. sphaericus* was also measured in another, closely situated Lake Peipsi [52], where both the *Microcystis* spp biomass and *mcyE* gene abundances in lake water are more than 10 times higher compared to Võrtsjärv [50,53,54]. This suggests that *C. sphaericus* feeding on toxigenic *Microcystis* is more likely connected to its individual demand for suitable food and not tightly influenced by the availability of *Microcystis* or the presence of toxin-producing cells in the surrounding lake water.

This result highlights an important finding regarding zooplankton feeding on toxigenic cyanobacteria in Võrtsjärv, and likely in other lakes with similar phytoplankton community composition. Even though *Microcystis* biomass and water column *mcyE* abundances might be low, reflecting an insignificant toxicity risk in the water body [55], the toxic cells are still effectively consumed by *C. sphaericus*, a species closely associated with these potentially toxic cyanobacteria. When preyed upon by invertebrates or fish, the toxic cells within the gut content of *Chydorus* are transferred to the next trophic levels in the food web [56] and can potentially cause problems for the food web despite insignificant levels of potentially toxic *Microcystis* in the lake water.

Current knowledge indicates that *C. sphaericus* is an important food object for planktivorous fish, such as young-of-the-year fish and bleak in Võrtsjärv [57]. The seasonal dynamics of *C. sphaericus* show substantial abundance and biomass decline during July–August, coinciding with the period of active feeding by young-of-the-year fish [37,58]. Although microcystin concentrations in the water column of Võrtsjärv are very low, current research results indicate that microcystin-producing cells can still be effectively transferred to higher levels in the food web via predation on *C. sphaericus*. We acknowledge the distinction between the presence of toxin production genes, actual toxin production, and potential toxin degradation [15] as it moves through the food web. Therefore, to confirm the proposed potential trophic transfer of microcystin via *Chydorus*, concentration measurements should be performed.

## 4. Conclusions

We found that the algal diet of the commonly prevailing small cladoceran in eutrophic waters, *Chydorus sphaericus*, consists predominantly of cyanobacteria, with a clear feeding preference for colonial forms. qPCR detection of *Microcystis*-specific *mcyE* genes confirmed the ingestion of toxigenic *Microcystis*. The results further suggest that, due to its affinity to colonial cyanobacteria, the trophic transfer of toxigenic *Microcystis* via *C. spaericus* can occur even in lakes with insignificant levels of potentially toxic *Microcystis* and under conditions of low water temperature, as demonstrated in Lake Võrtsjärv.

## 5. Materials and Methods

### 5.1. Study Site

Lake Võrtsjärv is a large lake in Estonia with an area of 270 km^2^ (58°05′–58°25′ N and 25°55′–26°10′ E). It is shallow (mean depth 2.8 m; maximum depth 6 m), with a high trophic state (total phosphorus~40 μg/L, total nitrogen~1000 μg/L, and chlorophyll *a* (Chl *a*) 32 μg/L), and high turbidity during the growing season with a Secchi depth less than 1 m [59,60]. The lake is ice-covered on average for 135 days per year, from the end of November to late April [59,60]. Cyanobacteria represent 60–95% of the total phytoplankton biomass [61,62], dominated by filamentous forms such as *Limnothrix planctonica* and *L. redekei*, accompanied by *Planktolyngbya limnetica* and *Aphanizomenon skujae*. The crustacean zooplankton biomass is dominated by small-sized cladocerans, including *C. sphaericus*, *Bosmina longirostris*, *Daphnia cucullata*, and juvenile cyclopoid copepods (mostly *Mesocyclops leuckarti*) [37,63]. Microcystins are present in the lake during the entire growing season, but the concentrations in the lake are very low, ranging from 20 to 120 ng/L [64].

### 5.2. Field Survey

Sample collection, along with environmental measurements, was performed monthly from January to December 2015 at the regular monitoring site, no. 10, in the deepest part of the lake [60]. On 18th of August, samples were collected from eight sampling sites (1, 2, 3, 4, 5, 6, 7, and 10) to compare feeding characteristics across the lake area (Figure 1). These sites correspond to plankton-dominated conditions, with cyanobacteria forming the majority of the phytoplankton biomass during the ice-free period [60]. However, between the sites, some compositional differences in phytoplankton and zooplankton communities occur, including variations in the abundance of *C. sphaericus*.

Water samples were collected at 1 m intervals with a Ruttner sampler from the entire water column and mixed in a tank to obtain a depth-integrated sample for analysis. Subsamples were taken from this pooled sample to analyze phytoplankton and zooplankton composition and biomass and to detect and quantify the phytoplankton pigments and potentially toxic cyanobacteria. For the zooplankton sample, 10 L of this water was filtered through a 48 μm plankton net. For phytoplankton, 200 mL was used. Phytoplankton and zooplankton samples were fixed with acidified Lugol’s iodine solution at a final concentration of 1% and preserved in the dark for further analysis. For molecular analysis, 100–500 mL (according to the sample density) of the depth-integrated water was filtered at low vacuum (max. 0.2 bar) through 5 µm pore size polycarbonate filters (Whatman, Cytiva, Marlborough, MA, USA). The samples were stored at −80 °C until analysis. To analyze phytoplankton marker pigments and potentially toxic cyanobacteria in *C. sphaericus* guts, depth-integrated samples of bulk zooplankton were collected from June to December with vertical tows of a 145 μm plankton net. The bulk zooplankton was instantly rinsed with deionized water to remove as much phytoplankton as possible, then concentrated in a small volume, and frozen in liquid nitrogen.

Background data (water temperature, pH, and dissolved oxygen) were measured with a YSI Professional multiprobe (Yellow Springs, OH, USA) concurrently with sampling, as a part of the national monitoring program. A Secchi disk was used to measure water transparency. Subsamples for water chemistry were analyzed by staff at the laboratory of the Estonian Environmental Research Centre, following international and Estonian quality standards (ISO and EVS-EN ISO).

### 5.3. Phyto- and Zooplankton Biomass and C. sphaericus Sample Preparations for Analysis

Phytoplankton samples were analyzed under an inverted microscope (Ceti Versus, Medline Scientific, Rotherham, UK) at ×400 magnification, following Utermöhl’s technique [65]. The counted taxa were converted to biovolumes by measuring cell/trichome/colony dimensions and approximating each taxon with a geometric shape. Zooplankton community composition and biomass were analyzed under a stereomicroscope (Nikon AZ100, Nikon Corporation, Tokyo, Japan) in a Bogorov chamber. For biomass, the lengths were converted to wet weight as described by [66,67,68]. Phytoplankton and zooplankton biomasses were expressed as mg WW/L (milligrams of wet weight per liter of lake water).

Before phytoplankton pigment and molecular analyses, frozen zooplankton samples were thawed to separate the *C. sphaericus* from other zooplankton species. From each subsample, 130 to 400 individuals were separated for phytoplankton pigment analysis, and 100 to 400 individuals were separated for molecular analysis. The separated specimens were rinsed with deionized water and inspected under a microscope to ensure no externally attached phytoplankton cells or filaments on the animals. For pigment analysis, *C. sphaericus* individuals were filtrated on GF/F filters (pore size 0.7 μm, Whatman, Cytiva, Marlborough, MA, USA), and for molecular analysis, they were placed into 1.5 mL microtubes for immediate DNA extraction.

### 5.4. Pigment Extraction and HPLC Analysis

In the present study, the affinity of the phytoplankton pigments is based on findings from the literature [69,70,71,72] and previously found correlations between water column pigments and phytoplankton groups in Lake Võrtsjärv [36]. Briefly, the marker carotenoids zeaxanthin and echinenone characterized cyanobacteria in general [69] while the carotenoid canthaxanthin was mainly associated with colonial cyanobacteria. Diatoms were represented by the carotenoids fucoxanthin and by the marker pigment group diadinoxanthin plus diatoxanthin [70]. Lutein, neoxanthin, and Chlorophyll *b* were used as proxies for chlorophytes. Alloxanthin represented cryptophytes [70,71]. Chlorophyll *a* is found in all primary producers and therefore represents the total phytoplankton biomass in this study [71].

The analysis of phytoplankton pigments followed slightly modified protocols from Leavitt and Hodgson [69] and Lie and Wong [73]. Depth-integrated water samples (100–300 mL) and rinsed zooplankton suspensions were filtered through Whatman GF/F 0.7 µm pore size filters and stored at −80 °C in the dark until phytoplankton pigment analysis. A solution of 90% acetone (by volume) with the internal standard (trans-β-apo-8′-carotenal, Sigma cat. #10810, Sigma-Aldrich, Darmstadt, Germany) was added to frozen GF/F filters to extract phytoplankton pigments. The samples were sonicated (Branson 1210, Branson Ultrasonics, Emerson Electric Co., St. Louis, MO, USA) for approximately 10 min in an ice bath under dim light and extracted at −20 °C in the dark for 24 h. Finally, the pigment extracts were clarified by filtration through a 0.45 µm filter (Millex LCR, Millipore MilliporeSigma, Darmstadt, Germany) before chromatographic analysis to remove any particles.

To separate phytoplankton pigments, reversed-phase high-performance liquid chromatography was applied, using a Shimadzu Prominence (Shimadzu Corporation, Kyoto, Japan) series binary gradient system with a photodiode-array and fluorescence detector. A fluorescence detector with an excitation wavelength set at 440 nm and emission at 660 nm was used to confirm the correct identification and low concentrations of Chl *a* [74]. Further details on the process of pigment analysis can be found in Tamm et al.’s study [75].

The estimated rates of pigment degradation in the zooplankton digestive system are rather controversial. Some studies demonstrate full preservation of pigments in the guts of copepods, cladocerans, and other small zooplankton [76,77], while others reveal significant degradation of carotenoids [78,79]. In this study, we assumed that Chesson’s selectivity index [80] represents a snapshot of phytoplankton pigments transferred to zooplankton. Considering the fast gut evacuation of small zooplankters [80], the pigment transfer is presumably much faster than the digestive changes affecting the carotenoid pigment composition [81,82].

### 5.5. DNA Extraction and Molecular Analysis

Genomic DNA from zooplankton was extracted using a DNeasy^®^ Blood and Tissue extraction kit (Qiagen Inc., Venlo, The Netherlands) following the manufacturer’s instructions. DNA from filtered water samples was extracted using a DNeasy^®^ PowerWater Kit (Qiagen Inc. Venlo, The Netherlands). All extractions were made under a laminar flow hood to protect samples and avoid contamination.

The quality and quantity of the extracted DNA were assessed using a NanoDrop 2000 UV-Vis spectrophotometer (Thermo Fisher Scientific Inc. Wilmington, DE, USA). The DNA was stored at −80 °C until further analysis. qPCR analyses were performed as described before [53,54,83]

Each environmental sample was tested in three replicates. Additionally, in every qPCR analysis, a negative control sample and positive standard DNA dilution series were included. qPCR reactions were conducted on a LightCycler^®^ 480 System (Roche Life Science, Roche Diagnostics, Basel, Switzerland) using a 384-well platform using the following protocol: 95 °C for 12 min for initial denaturation, 40 cycles of 95 °C for 15 s, and 62 °C. The results were analyzed using LightCycler^®^ Software 1.5. The *mcyE* gene was chosen to detect and quantify potential microcystin-producing *Microcystis* because of its established role in microcystin production and its reliability as a molecular marker. Since *mcyE* is typically found as a single copy per genome, it is ideal for assessing the abundance of potentially toxic *Microcystis* cells in both environmental samples and grazers [54,84,85,86].

### 5.6. Data Analysis

To assess the feeding selectivity of zooplankton, the alpha selectivity index of Chesson [80] was used. The formula used for calculations is as follows:$$\alpha_{i}=\frac{r_{i}/p_{i}}{\sum_{i=1}^{n}r_{i}/p_{i}};i=1\ldots n$$
where *r_i_* is the percentage of *i*-th phytoplankton pigments in zooplankter guts, *p*_i_ is the percentage of the same phytoplankton pigments in the lake water, and *n* is the total number of pigments analyzed. When a = 1/*n* (in the present study 1/*n* = 0.111), zooplankton feeding is non-selective. Values of *α_i_* > 0.111 or *α*_i_ < 0.111 indicate selection and avoidance of carotenoid pigments and respective phytoplankton groups by zooplankton, respectively.

To test the significance of relationships with toxigenic cyanobacteria, the nonparametric Kruskal–Wallis test was first used to evaluate the differences between sampling sites in *mcyE* copy numbers in lake water and *C. sphaericus* ingestion of *Microcystis* containing *mcyE* genes. Spearman’s rank correlation (*rs*) with the function “cor.test” was used to assess the relationship between *mcyE*-containing cells in *C. sphaericus* gut content and *Microcystis mcyE* cells in water, as well as phytoplankton indices. The relationship between *Microcystis mcyE*-containing cells in water and physicochemical, phytoplankton, and zooplankton indices (including *Chydorus*, cladoceran abundances and biomass, and total metazooplankton biomass) was also tested. Spearman’s rank correlation analyses were performed using the RStudio 4.1.2 package and its extensions.

## Figures and Tables

**Figure 1 toxins-17-00298-f001:**
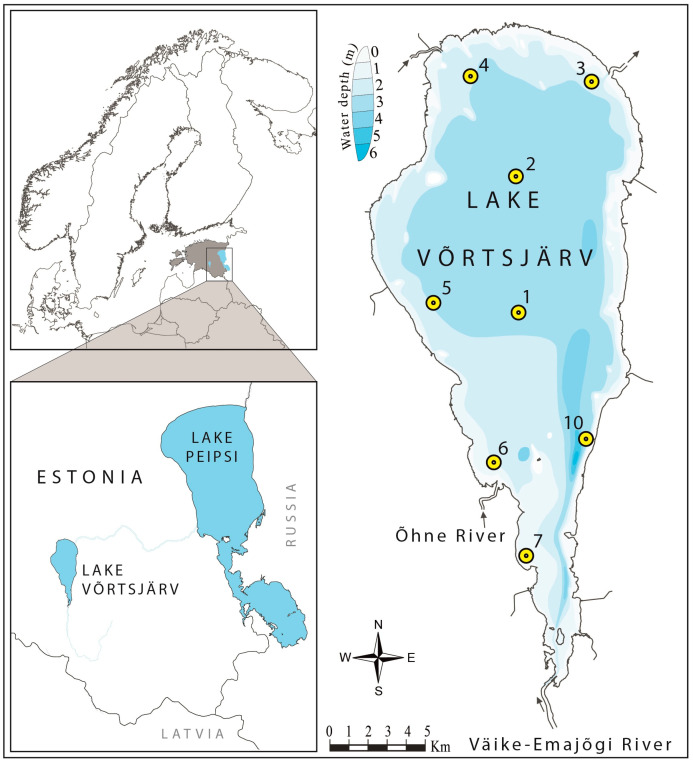
Locations of sampling sites in Lake Võrtsjärv.

**Figure 2 toxins-17-00298-f002:**
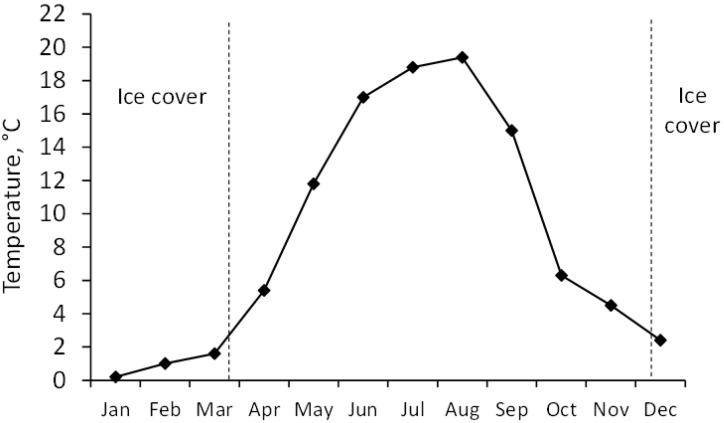
Temperature dynamics at sampling site 10 in Lake Võrtsjärv in 2015.

**Figure 3 toxins-17-00298-f003:**
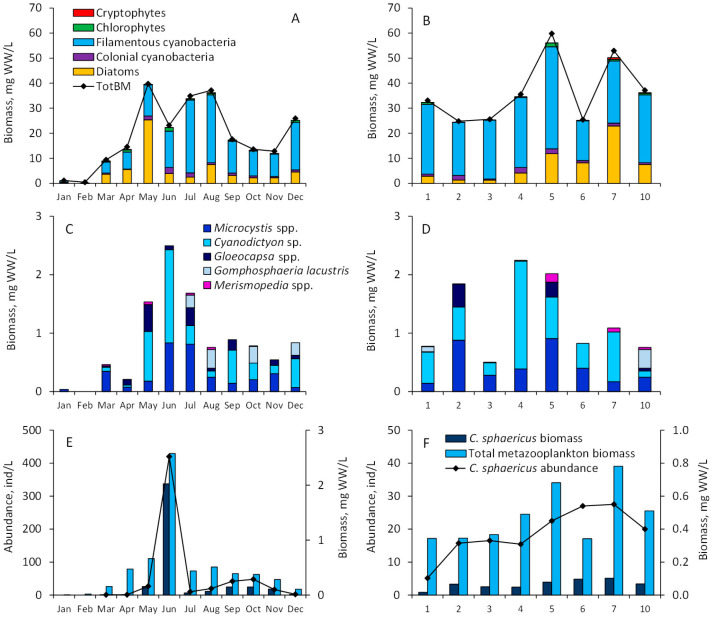
Seasonal dynamics of major phytoplankton groups and total phytoplankton biomass (TotBM) at sampling site 10 (**A**) and at sites sampled in August across Lake Võrtsjärv (**B**). Seasonal dynamics and composition of colonial cyanobacteria at sampling site 10 (**C**) and at sites sampled in August across the lake (**D**). Seasonal abundance and biomass of *C. sphaericus* and total zooplankton at sampling site 10 (**E**) and at sites sampled in August across the lake (**F**).

**Figure 4 toxins-17-00298-f004:**
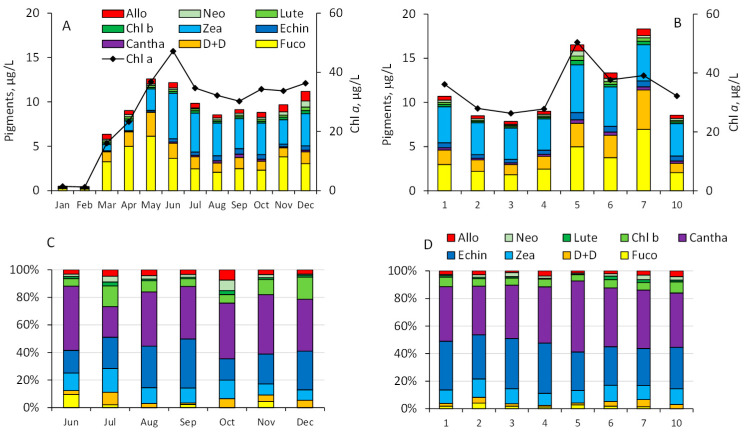
Seasonal dynamics of phytoplankton pigments in depth-integrated water samples from sampling site 10 (**A**), and at sites sampled in August in Lake Võrtsjärv (**B**). Percentage contribution of phytoplankton pigments in the guts of *C. sphaericus* at sampling site 10 (**C**) and at sites sampled across the lake (**D**); Chl *a*—chlorophyll *a*; Cantha—canthaxantin; Zea—zeaxanthin; Echin—echinenone; Fuco—fucoxanthin; D+D—diadinoxanthin+diatoxanthin; Lut—lutein; Chl *b*—chlorophyll *b*; Allo—alloxanthin; Peri—peridinin.

**Figure 5 toxins-17-00298-f005:**
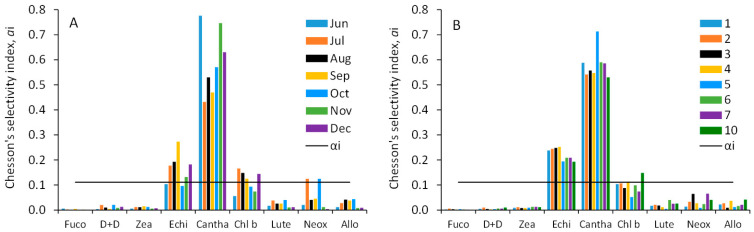
Chesson’s selectivity index of C. sphaericus for phytoplankton pigments in Lake Võrtsjärv at sampling site 10 (**A**) and at sites sampled in August across the lake (**B**); Values above the horizontal line (αi > 0.111) indicate positive selection. Fuco—fucoxanthin; D+D—diadinoxanthin+diatoxanthin; Zea—zeaxanthin; Cantha—canthaxantin; Echin—echinenone; Lut—lutein; Chl b—Chlorophyll b; Allo—alloxanthin; Peri—peridenin.

**Figure 6 toxins-17-00298-f006:**
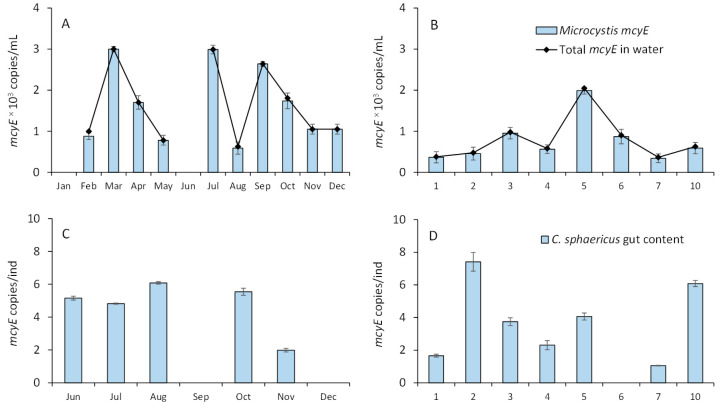
Seasonal dynamics of *mcyE* synthase genes at sampling site 10 (**A**) and at sites sampled in Lake Võrtsjärv in August (**B**). The abundance of *Microcystis mcyE*-containing cells in the gut content of *C. sphaericus* at sampling site 10 (**C**) and at the sampling sites analyzed in Lake Võrtsjärv in August (**D**). Whiskers represent 2SD from the mean, and bars represent mean values.

## Data Availability

The original contributions presented in this study are included in the article/Appendix A. Further inquiries can be directed to the corresponding author.

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
