# Peer review of "Cladoceran Chydorus sphaericus and Colonial Cyanobacteria: Potentially a Toxic Relationship?"

_toxins, 2025, doi:10.3390/toxins17060298_

Round 1
Reviewer 1 Report
Comments and Suggestions for Authors
I mainly have three comments below.
- Figure 3C and 3D - the blue shades are quite close for different species.
- Figure 6A and 6B - "total mcyE" should be mcyE in water samples, correct? How did the experiment separate the microcystis and total water samples? Was there a filtration step before DNA extraction?
- Most figures presented seem to only have a single measurement. Are there replicates measured? The number of samples/replicates should be added.
Reviewer 2 Report
Comments and Suggestions for Authors
As the authors correctly observed, previous work has left open the questions of whether smaller bloom-adapted zooplankton cost-effectively feed on toxic cyanobacteria and whether, in doing so or not, they select for the toxic cyanobacteria. These questions are important to answer so that we can better understand the spatial and temporal distributions of toxic blooms and their effects on food web dynamics. The present study has helped to do this by demonstrating that C. sphaericus can and does preferentially ingest toxic colonial cyanobacteria but not at a consistently density-dependent rate. It also demonstrates the occurrence of trophic transfer of toxic cyanobacteria from C. sphaericus gut content. Overall, I found the study well thought-out and the manuscript well-written.
However, I believe that the authors need to somehow (speculatively, at least) broach the subject of why toxic cyanobacterial blooms occur in the presence of C. sphaericus (and other small toxin-tolerant zooplankton), given that the authors have shown that these grazers feed on toxic cyanobacteria even when they are at low density at the start of the growth season and are not deterred by the toxins even when cyanobacterial abundance is at its peak. This is relevant not only because of the widespread interest in using zooplankton as potential biological control agents against toxic cyanobacteria but also because the answers might clarify one of the results of the study that I personally found confusing.
The result I refer to is reported in Lines 192-202 (and rephrased in LINES 259-261). There were no statistical differences among lake water, phytoplankton biomass, and zooplankton gut content in mycE measures, but also no correlation between mycE and the subsequently described parameters (apart from temperature). The data could perhaps be too scarce or noisy for the statistical approach used, but they do appear so to me, which suggests that the lack of difference can be interpreted as similarity (which, in turn, should be reflected in the correlations). I can imagine toxic cyanobacterial abundance in the guts being decoupled with toxic cyanobacterial abundance in the habitat if the cyanobacteria are digested at a rate much faster or slower than the rate at which they proliferate (creating time lags). I can also imagine it occurring if higher trophic levels (e.g., fish) control zooplankton abundance with variable effects on the zooplankton’s per capita consumption of cyanobacteria or if the cyanobacteria happen to be symbiotic gut microflora of the zooplankton.
Additionally, I noticed some proofreading issues the authors should address. These are as follows:
- Line 30: Microcystis should be followed by “sp.” or “spp.” (as should Paramaecium in Line 39), as other genera are elsewhere in the manuscript (also, “sp.” and “spp.” should consistently have a period to indicate their status as abbreviations).
- Line 35: “has” should be “have.”
- Line 39: while “zooplankters” is technically correct, it would be better to use “zooplankton” as the plural consistently throughout the manuscript.
- Line 41: the “of” in “of survival and fecundity” should be deleted.
- Line 43: given the taxonomic depth of the descriptions up to this point, I would replace “algae” with “phytoplankton” to avoid confusion.
- Line 65: the “as” is missing. Also, “s.l.” should be deleted, since the authors do not actually use the sensu lato abbreviation when referring to their specimens.
- Lines 224-225: It looks two different draft sentences have been accidentally mixed and matched here to create incomplete thoughts. One or the other should be selected and completed.
- Line 261: “in surrounding” should be “in the surroundings” or “in surrounding lake water."
Reviewer 3 Report
Comments and Suggestions for Authors
I read the manuscript and got two major issues which i could not find any explanation.
- The first issue- is that pigments used as diagnostic (proxy) for cyanobacteria are not unique to cyanobacteria. Zeaxanthin for example is found in majority of phptotrophic organisms. Unfortunately i could not find the text for the paper which lies as the foundation for the method.
- Second issue, there is not data for actual toxin production. The presence of the gene is one thing, but is there toxin release? Is it bioaccumulated in the zooplankton? These points are not demonstrated.
I do appreciate the amount of work being done. I do not work with environmental and ecosystems issues, and it may happen, that the methods used here are now widely accepted in the field. But it was not convincing enough for me.
Round 2
Reviewer 1 Report
Comments and Suggestions for Authors
All my comments have been addressed.